# Juvenile Hormone Synthesis Pathway Gene *SfIPPI* Regulates *Sogatella furcifera* Reproduction

**DOI:** 10.3390/insects13020174

**Published:** 2022-02-06

**Authors:** Ming-Fu Gong, Xi-Bin Yang, Gui-Yun Long, Ze-Yan Jia, Qing-Hui Zeng, Dao-Chao Jin, Hong Yang, Cao Zhou

**Affiliations:** 1Provincial Key Laboratory for Agricultural Pest Management of Mountainous Region, Institute of Entomology, Guizhou University, Guiyang 550025, China; gongmingfu2019@126.com (M.-F.G.); yangxibingz@126.com (X.-B.Y.); lgy0256@126.com (G.-Y.L.); Jiazeyan520@163.com (Z.-Y.J.); 18311516405@163.com (Q.-H.Z.); daochaojin@126.com (D.-C.J.); 2College of Life Sciences, Chongqing Normal University, Chongqing 401331, China; zhouc@cqnu.edu.cn

**Keywords:** *Sogatella furcifera*, juvenile hormone, synthesis pathway, isopentenyl pyrophosphate isomerase (IPPI), RNA interference, reproduction

## Abstract

**Simple Summary:**

The juvenile hormone is essential for insect growth, development, and reproduction. Isopentenyl pyrophosphate isomerase is a key isomerase involved in the synthesis of the juvenile hormone. This study evaluates the effect of the targeted silencing of the *SfIPPI* gene on the reproduction of *Sogatella furcifera* (white-backed planthopper). We found that *SfIPPI* silencing significantly inhibits the ovarian development and egg production in female adults of *S. furcifera* and significantly inhibits the transcription of downstream genes in the juvenile hormone synthesis pathway. Our data provide insight into the function of juvenile hormone biosynthetic pathway genes in insect reproduction, which could be a potential target to control and prevent agricultural pests.

**Abstract:**

The juvenile hormone (JH) is crucial for insect reproduction, and isopentenyl pyrophosphate isomerase (IPPI) is a key enzyme in the JH synthesis pathway. However, few studies have investigated how IPPI regulates insect reproduction. This study identifies and characterizes the IPPI gene (*SfIPPI*) from the important agricultural pest *Sogatella furcifera*. A phylogenetic analysis reveals a high homology of *SfIPPI* with the *IPPI* amino acid sequences of *Laodelphax striatellus* and *Nilaparvata lugens* (Stål). Furthermore, *SfIPPI* is expressed at various developmental stages and in various tissues of *S. furcifera,* and is significantly higher on the 5th day of adult emergence and in integument tissue, while lower levels are found on the 3rd day of adult emergence and in fat body and gut tissue. After silencing *SfIPPI* using RNA interference, the ovarian development is significantly inhibited and the fecundity is significantly reduced when compared with the control group. Additionally, *SfIPPI* silencing significantly decreases the expression levels of downstream JH signal transduction pathway genes (*SfJHAMT*, *SfFAMeT*, and *SfKr-h1*) and *SfVg*. Our findings are helpful in elucidating the molecular mechanism underlying the regulation of insect reproduction through genes in the JH synthesis pathway, and they provide a theoretical basis for the development of pest control treatments targeting *SfIPPI*.

## 1. Introduction

The juvenile hormone (JH) is a sesquiterpenoid compound synthesized in the corpora allata of insects [1,2]. It is essential for regulating the physiological processes of insect growth, metamorphosis, reproduction, and immunity [3,4,5,6,7,8]. JH biosynthesis is associated with the expression of most JH biosynthetic enzymes in the corpora allata [9]. Isopentenyl pyrophosphate isomerase (IPPI) is an isomerase involved in JH synthesis [10,11] that catalyzes the conversion of isopentenyl pyrophosphate (IPP) to dimethylpropylene pyrophosphate (DMAPP). IPPIs were first extracted from the lepidopteran insect *Bombyx mori* in 1985 [12], and the *IPPI* sequence was, subsequently, cloned from insects such as *Choristoneura fumiferana* and *Manduca sexta* [13]. Later, a study of the molecular characteristics and function of *Aedes aegypti AaIPPI* revealed that it was important for JH synthesis because changes in the level of *AaIPPI* mRNA affected JH biosynthesis [14]. However, there are relatively few studies investigating how IPPIs regulate insect reproduction.

The white-backed planthopper, *Sogatella furcifera* (Horváth) (Hemiptera: Delphacidae), is an important insect pest in rice production in China and Asia [15]. *S. furcifera* completes its generational development with the use of rice crops and weeds, causing extensive crop damage. It is characterized by a short life cycle, long-distance migration, and strong reproductive and environmental adaptability [16,17]. Most of the crop damage is caused by adults and nymphs directly sucking the phloem sap of rice plants and by females laying eggs. *S. furcifera* is also a transmission vector for the southern rice black-streaked dwarf virus [18], which negatively affects rice production and leads to severe economic losses [19,20,21]. At present, *S. furcifera* is mainly controlled using chemical insecticides, but their excessive use not only negatively impacts the environment and natural enemies of *S. furcifera,* but also lead to insecticide resistance and other problems [22,23,24]. Thus, to achieve a sustainable pest management, it is imperative to find an environmentally friendly and efficient method to control *S. furcifera* populations.

Insects have a strong reproductive ability, and studies of the regulation of their reproductive mechanisms are crucial for pest control. Significant work has been conducted on the role of the JH in the regulation of reproduction in the brown planthopper, *N. lugens*. Previous studies have shown that JH analogues can stimulate vitellogenesis in this species [25]. An adequate nutrition has been found to affect JH biosynthesis and, thus, reproductive maturation [26,27]. Additionally, the loss of brummer-mediated lipolysis was found to impair vitellogenesis and oocyte maturation by working through the JH signaling pathway [28]. In this study, we clone and identify *SfIPPI* based on the published genome [29] and transcriptome data [30] of *S. furcifera*. We used quantitative (q) polymerase chain reaction (PCR) assays to detect the expression levels of *SfIPPI* in different developmental stages and tissues, and also used RNA interference (RNAi) technology to study the function of this gene in *S. furcifera* reproduction. Our findings provide a reference for the sustainable management of *S. furcifera* by identifying suitable target genes to control the rapid propagation of these pests.

## 2. Materials and Methods

### 2.1. Insect Rearing and Collection

*S. furcifera* were collected from a rice field in Huaxi, Guiyang, Guizhou, China, in 2013. They were normally reared and reproduced on Taichung Native 1 rice in our laboratory under specific environmental conditions (temperature, 25 ± 1 °C; relative humidity 70% ± 5%; photoperiod, 16 h light: 8 h dark). They were kept in an isolated area so that the adults could be used as the source of test insects.

### 2.2. Sample Preparation

We collected and prepared samples of adult *S. furcifera* to determine the expression levels of *SfIPPI* in different developmental stages and tissues. Firstly, we paired and reared newly emerged females and males of *S. furcifera* in test tubes with rice seedlings that were replaced daily. During the feeding process, we randomly collected samples from egg to adult at 16 time points: egg (*n* = 40); 1st (*n* = 40); 2nd (*n* = 40); 3rd (*n* = 35); 4th at 1d and 2d (*n* = 30); 5th at 1d, 2d, and 3d (*n* = 20) and female adults at 12 h, 1 d, 2 d, 3 d, 4d, and 5d after eclosion (*n* = 15) (*n* = 15–40 individuals per replicate, respectively). Secondly, we dissected the head (*n* = 50), gut (*n* = 100), fat body (*n* = 50), ovary (*n* = 30), and integument (*n* = 100) tissues from 24 h old female adults in phosphate-buffered saline (PBS; pH = 7.4). All samples were subjected to three biological replicates. Each collected sample was immediately frozen in liquid nitrogen and placed in a 1.5 mL RNAse-free microcentrifuge tube before storing at −80 °C until RNA extraction.

### 2.3. Total RNA Extraction and cDNA Synthesis

We extracted total RNA from samples collected at different developmental stages and tissues of *S. furcifera* using an EZNA HP Total RNA Kit (Omega Bio-tek, Norcross, GA, USA) according to the manufacturer’s recommendations. The concentration of the extracted RNA was determined using a NanoDrop 2000 spectrophotometer (Thermo Fisher Scientific, Waltham, MA, USA). The RNA integrity was evaluated following 1% agarose gel electrophoresis. Then, the extracted qualified RNA underwent reverse transcription using a PrimeScript RT Reagent Kit with gDNA Eraser (TaKaRa, Dalian, China) to synthesize first-strand cDNA, which was stored at −20 °C until use.

### 2.4. SfIPPI Cloning

Based on the published genome [29] and transcriptome [30] data of *S. furcifera*, we searched for the cDNA sequence of *SfIPPI* using the BLAST tool (https://blast.ncbi.nlm.nih.gov/Blast.cgi, accessed on 26 May 2021) on the NCBI website. Following alignment of the obtained sequences, we designed primers for the open reading frame (ORF) sequence of *SfIPPI* (Table 1) using Primer Premier 6.0 software (PREMIER Biosoft International, Palo Alto, CA, USA). Then, we performed PCR assays using the first-strand cDNA that we already obtained by reverse transcription as the template. The PCR reaction mixture included 12.5 μL PCR master mix, 1 μL each of the forward and reverse primers, and 3 μL *S. furcifera* cDNA template, which was composed to a final volume of 25 μL using ddH_2_O. The cycling conditions were as follows: pre-denaturation at 94 °C for 3 min; 30 cycles of denaturation at 94 °C for 30 s, annealing at 55 °C for 30 s, and extension at 72 °C for 1 min; final extension at 72 °C for 10 min. The PCR products were stored at 4 °C and evaluated following 1% agarose gel electrophoresis. Then, the target band was excised from the gel and purified using an EasyPure Quick Gel Extraction Kit (Quanshijin Biotechnology, Beijing, China) according to the manufacturer’s instructions. The recovered PCR product was ligated to a pMD18-T vector (TaKaRa, Dalian, China) and transformed into competent *Escherichia*
*coli* DH5α cells (Quanshijin Biotechnology, Beijing, China). Finally, possible colonies containing the vector were sent to Sangon Biotech (Shanghai, China) for sequencing.

### 2.5. Sequencing Analysis

After using SeqMan software (DNASTAR, Madison, WI, USA) to assemble and proofread the sequencing data, we used DNAMAN 7.0 software (Lynnon Biosoft, San Ramon, CA, USA) to deduce the amino acid sequence. We also used the BLAST tool on the NCBI website to further confirm the target gene and obtain the homologous sequences. We used the ORF Finder tool (https://www.ncbi.nlm.nih.gov/orffinder/, accessed on 27 May 2021) on the NCBI website to identify the ORFs of *SfIPPI*; ProtParam (https://web.Expasy.org/protparam/, accessed on 27 May 2021) to predict the amino acid molecular composition, relative molecular mass, isoelectric point, and other physical and chemical properties of the encoded protein; the SMART database (http://smart.embl-heidelberg.de/, accessed on 27 May 2021) to predict conserved domains. We constructed a phylogenetic tree using the neighbor-joining method in MEGA 6.06 software [31] with 1000 bootstrap replications.

### 2.6. SfIPPI Expression Different Developmental Stages and Tissues

We used Primer Premier 6.0 software to design qPCR primers (Table 1) based on the *SfIPPI* sequence obtained from the previous cloning and identification experiments. qPCR analysis was performed on a CFX96 Real-Time PCR System (Bio-Rad, Hercules, CA, USA) using FastStart Essential DNA Green Master hot-start reaction mix (Roche, Indianapolis, IN, USA) to detect the expression levels of *SfIPPI* in the different developmental stages and tissues of *S. furcifera*. The qPCR reaction was performed in a final volume of 20 μL containing 1 µL sample cDNA, 1 µL of each forward and reverse primer (10 µM), 7 µL RNase-free water, and 10 µL FastStart Essential DNA Green Master. The qPCR cycling parameters were initially denatured at 95 °C for 10 min, followed by 40 cycles of amplification (95 °C for 30 s and 60 °C for 30 s). Melting curve analysis was performed from 65 °C to 95 °C. The ribosomal protein L9 (GenBank accession number: KM885285) and α-1 tubulin (GenBank accession number: KP735521) genes were used as internal controls. Three biological replicates and three technical replicates were performed for each developmental stage and tissue.

### 2.7. Double-Stranded RNA Synthesis

Based on the previously obtained *SfIPPI* sequence, we designed gene-specific primers containing the T7 polymerase promoter sequence at the 5ʹ-terminal using the E-RNAi website, an online tool for the design and evaluation of RNAi reagents. (https://www.dkfz.de/signaling/e-rnai3/, accessed on 12 June 2021) (Table 1). Then, we used these primers for PCR amplification of our previously synthesized first-strand cDNA to generate a DNA template containing the T7 promoter sequence. Next, we used TA cloning to expand and cultivate the population of transformed bacteria in a liquid medium. Then, the plasmid was extracted, and the high-concentration gel-recovered dsDNA was used as the template for dsRNA synthesis. The dsRNA was synthesized using a TranscriptAid T7 High-Yield Transcription Kit and GeneJET RNA Purification Kit (both Thermo Fisher Scientific) according to the manufacturer’s instructions. The concentration of the dsRNA was determined using a NanoDrop 2000 spectrophotometer, and its integrity was evaluated following 1% agarose gel electrophoresis. At the same time, the green fluorescent protein (GFP) gene was used to synthesize GFP dsRNA by the same method for use in the control group.

### 2.8. RNAi Experiment

We performed an RNAi experiment to investigate the function of *SfIPPI* in *S. furcifera* reproduction. Newly emerged (1–12 h) female adults of *S. furcifera* were selected as the test insects in this study. Following anesthetization with CO_2_ for 30 s, we placed the test insects in empty culture plates. Then, we used an IM-31 microinjector (NARISHIGE, Tokyo, Japan) to inject 100 nL ds*IPPI* into the thorax between the middle and hind leg. An equal volume of ds*GFP* was injected in the negative control group. Three biological replicates were performed for each treatment, with 100 female adult injections for each replicate. The injected insects were placed in a test tube containing fresh rice seedlings and kept in an artificial climate chamber (temperature; 25 ± 1 °C; relative humidity, 70 ± 5%, and photoperiod, 16 h light: 8 h dark) for 48 h. Then, 10 of the surviving *S. furcifera* were randomly selected from each experimental group to determine RNAi efficiency using reverse transcription (RT)-qPCR. At the same time, the transcription levels of *SfVg* and *SfVgR* genes and JH synthesis pathway-related genes were determined to clarify their regulatory relationship. Three biological replicates were performed in this experiment, and 10 injected adults were used for each biological replicate.

### 2.9. Bioassay

To assess the effect of RNAi on the fertility of *S. furcifera*, we injected emerged females with ds*IPPI* or ds*GFP* (as the control). Then, one female and two males (two uninjected males, excluding the influence of males on mating) were paired after injection in glass tubes containing fresh rice seedlings. Each treatment group contained 15 insect parings, and three biological replicates were performed. Fresh rice seedlings were provided every 2 days, and the old seedlings were kept in the tube for further culture and observation. The number of newly hatched nymphs was recorded daily. After 10 days, the old rice seedlings were dissected under a stereomicroscope, and the number of hatched nymphs and unhatched eggs were also counted until the female adults died. Ovaries were dissected 6 days after injection and washed with PBS before using a stereoscopic microscope (SMZ25 Nikon Corporation, Tokyo, Japan) to observe, compare, and photograph ovarian development.

### 2.10. Statistical Analysis

The 2^−ΔΔCt^ method [32] was used to calculate the relative expression of *SfIPPI* and other genes in the JH synthesis pathway. All experimental data were analyzed using SPSS 22.0 statistical software (IBM Corp., Armonk, NY, USA). Significant differences between the treatments were assessed using one-way analysis of variance, followed by the Tukey test for multiple comparisons. The Student *t*-test for independent samples was used to analyze the significance of the efficiency of RNAi silencing.

## 3. Results

### 3.1. Sequence Identification and Characteristics of SfIPPI

According to the verification results, the final full-length ORF of *SfIPPI* from *S. furcifera* contained 633 bp that encoded a hypothetical protein sequence of 210 amino acids. After BLAST alignment, we named it *SfIPPI* (GenBank accession number: OM417142) according to its sequence similarity and conserved domains. The online software ProtParam predicted the molecular formula of the *S. furcifera* IPPI protein as C_1090_H_1680_N_310_O_312_S_9_ with a molecular weight of approximately 24.41 kDa and a theoretical isoelectric point of 6.30. The conserved domain analysis revealed that the protein contained a typical conserved domain of the Nudix family (amino acid positions 71–102), a conserved cysteine and conserved glutamic acid motifs, and seven conserved domains that are key to the catalytic activity of IPPI enzymes (Figure 1).

### 3.2. Sequence Comparisons and Phylogenetic Analysis

We investigated the degree of similarity between the IPPI protein of *S. furcifera* and that of other species using a BLAST homology search and comparison analysis. *SfIPPI* had high similarities of 92.38% and 85.24% with *Laodelphax striatellus* (GenBank accession number: RZF47560) and *Nilaparvata lugens* (Stål) (GenBank accession number: XP_022187401), respectively. Furthermore, there was a 57.14% similarity with the hemipteran *Halyomorpha halys* (GenBank accession number: XP_014271459). The phylogenetic tree (Figure 2) showed the *S. furcifera* IPPI protein clustered with *N. lugens*, *L. striatellus*, *H.*
*halys*, and *Riptortus pedestris*, forming a branch comprising Hemiptera. However, the *S. furcifera* IPPI protein was most closely related to that of *N. lugens* and *L. striatellus*.

### 3.3. SfIPPI Expression in Different Developmental Stages and Tissues

The expression pattern of *SfIPPI* in the different developmental stages of *S. furcifera* was detected by qPCR. *SfIPPI* was expressed and fluctuated in all developmental stages of *S. furcifera*. The expression level was higher in the third instar and fourth instar 1 d old nymphs and in emerged female adults at 12 h, 1 d, and 5 d. The lowest expression level was in 3 d old female adults (Figure 3A). *SfIPPI* was expressed in various tissues of *S. furcifera* adults. The levels were highest in the integument and head, followed by the ovary, and lowest in the gut and fat body, with the fat body showing the lowest level of expression (Figure 3B).

### 3.4. Effect of Silencing of SfIPPI on Reproduction of S. furcifera Female Adults

We determined the level of *SfIPPI* mRNA 48 h after injecting newly emerged females with ds*IPPI* of *S. furcifera*. Compared with the control group (injected with ds*GFP*), the expression of *SfIPPI* was significantly suppressed and downregulated by 66.35% (Figure 4A), indicating that *SfIPPI* was successfully silenced. Furthermore, the number of females laying eggs was reduced to 73.37% in the ds*IPPI* group, which was significantly lower than the control group (219 eggs) (Figure 4B), indicating that *SfIPPI* significantly affected the number of eggs laid by *S. furcifera*. To further understand the effect of *SfIPPI* on the ovarian development of *S. furcifera*, we dissected the ovaries 6 d after injection to observe their development (Figure 4C). We observed a large number of typical banana-shaped mature oocytes in the ds*GFP* group, indicating that ds*GFP* injection did not affect ovarian development. However, a few mature oocytes were observed in the ds*IPPI* treatment group, suggesting that *SfIPPI* significantly affected ovarian development.

### 3.5. Effect SfIPPI Silencing on Other Genes in the JH Signal Transduction Pathway

It has been reported that the JH signaling pathway gene and vitellogenin gene can regulate insect reproduction. In order to clarify the mechanism of the effect of *SfIPPI* gene silencing on the reproduction of *S. furcifera,* we detected changes in the expression of genes related to the JH signal transduction pathway of *S. furcifera* in newly emerged females, following injection with ds*IPPI*. The expression levels of *SfJHAMT* and *SfFAMeT* and those of *SfKr-h1* and *SfVg* were significantly lower compared to the control group (*p* < 0.05 and *p* < 0.01, respectively). The expression levels of *SfMet* and *SfVgR* did not change significantly (Figure 5).

## 4. Discussion

*IPPI* is a crucial enzyme in the mevalonate pathway in JH synthesis, which catalyzes the conversion of IPP to DMAPP [11] and is crucial for JH biosynthesis. In this study, we successfully cloned the ORF sequence of *IPPI* from *S. furcifera* using PCR. Bioinformatic analyses revealed that, similar to other insects, *SfIPPI* encoded a protein sequence containing a typical conserved domain of the Nudix family. Additionally, the phylogenetic analysis showed that *SfIPPI* was highly similar to *L. striatellus* and *N. lugens*.

The expression levels of *SfIPPI* in different developmental stages and tissues of *S. furcifera* were detected by qPCR. *SfIPPI* was expressed and fluctuated in all developmental stages of *S. furcifera* and was highest in 5 d old adults. *SfIPPI* expression was detected in all tissues and was highly expressed not only in the integument and head tissues, but also in ovary tissue, suggesting that IPPI is essential for the reproduction of adult *S. furcifera*. The *IPPI* expression levels were similar to those of other insects. In *A.*
*aegypti*, the mRNA level of *AaIPPI* in the corpus allatum was reported to increase 6 d before eclosion and was highest 24 h after eclosion, and changes in the pattern of JH biosynthesis were similar [14]. Furthermore, a study of *Dendroctonus armandi* revealed that *DaIDI* expression increased significantly during the adult stage [33]. Similarly, during the development of *Leptinotarsa decemlineata*, the expression level of *LdIDI* fluctuated during development, and *LdIDI* expression was detected in the midgut, Malpighian tube, fat body, epidermis, and other tissues [34].

Many studies have investigated how other genes in the JH signal transduction pathway regulate insect reproduction. For example, after RNAi silencing, the expression of *BdHMGR*, *BdMet*, and *BdKr-h1* and the ovary development of the female *Bactrocera dorsalis* was significantly inhibited, and the amount of oviposition was significantly reduced [35,36]. In *N. lugens*, after RNAi of JH signaling pathway genes *NlTai*, *NlMet*, and *NlKr-h1*, ovarian development was affected, the pre-oviposition period was prolonged, and the number of oviposited eggs was reduced [37,38,39]. Similarly, the RNAi inhibition of the *FAMeT* gene in males or females significantly reduced the number of eggs laid by female *Ceratitis capitata* [40]. Furthermore, Gijbels et al. [41] reported that female *Schistocerca gregaria* injected with ds*Met* could not lay eggs.

All of the above studies have confirmed that JH signal transduction pathway genes are essential for insect reproduction. The inhibition of JH signal transduction genes by RNAi, subsequently, leads to the inhibition of ovarian development, thereby reducing the amount of oviposition and prolonging preoviposition. However, there are few reports of the regulation of insect reproduction by IPPIs. In this study, we used RNAi technology to analyze the role of the JH synthesis pathway gene *SfIPPI* in *S. furcifera* reproduction. Our results showed that the ovarian development of females treated with ds*IPPI* was significantly inhibited and the number of eggs laid was significantly lower compared to the control group (treated with ds*GFP*), which was consistent with the function of other genes in the JH synthesis pathway. Targeted silencing of *SfJHAMT* and *SfFAMeT* of *S. furcifera* inhibited the fecundity, ovarian development, and transcription levels of *SfKr-h1* and *SfVg* significantly [42]. These findings suggest that *SfIPPI* affects the ovarian development of *S. furcifera* and, consequently, the number of eggs laid.

In this study, we showed that JH synthesis pathway genes are crucial for insect reproduction. Furthermore, *SfIPPI* silencing significantly inhibited the transcription levels of JH signal transduction pathway-related genes (*SfJHAMT*, *SfFAMeT*, and *SfKr-h1*) and *SfVg* in *S. furcifera*. Our findings suggest that *SfIPPI* affects the ovarian development and fecundity of *S. furcifera* by regulating the transcription levels of other genes in the JH signal transduction pathway and of *SfVg*. Similar results have been observed in the cotton bollworm (*Helicoverpa armigera*), where the targeted silencing of *HMGR* expression significantly inhibited female oviposition and *Vg* expression [43]. Thus, *the* JH biosynthetic pathway gene *SfIPPI* plays a crucial role in insect ovarian development and reproduction.

These results can provide a theoretical basis for the application of JH in the green control of rice planthoppers.

## 5. Conclusions

In summary, we identified *SfIPPI* in the *S. furcifera* genome and transcriptome. The bioinformatics analysis revealed a typical conserved domain of the Nudix family, and the phylogenetic analysis showed that *SfIPPI* had the highest similarity with *L. striatellus* and *N. lugens*. The pattern expression analysis indicated that *SfIPPI* was expressed in all developmental stages and in different tissues of *S. furcifera*, which may be associated with physiological processes. Additionally, *SfIPPI* silencing significantly inhibited the ovarian development and fecundity of females and significantly inhibited the transcription level of downstream genes in the JH synthesis pathway. Our data could help to clarify the structure, phylogeny, expression models, and biological functions of *SfIPPI* to further understand the function of JH biosynthetic pathway genes in insect reproduction and to provide a reference for the prevention and control of agricultural pests in the future.

## Figures and Tables

**Figure 1 insects-13-00174-f001:**
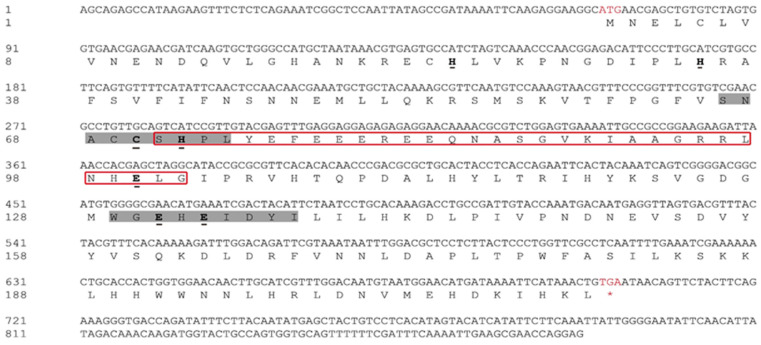
Nucleotide and amino acid sequence analysis of *SfIPPI* in *S. furcifera*. Red font ATG, start codon; red font TGA with an asterisk, stop codon; red box, Nudix hydrolase domain; bold and underlined amino acids, residues critical for the catalytic activity of the enzyme; amino acids with a shaded background, conserved cysteine and conserved glutamic acid motifs.

**Figure 2 insects-13-00174-f002:**
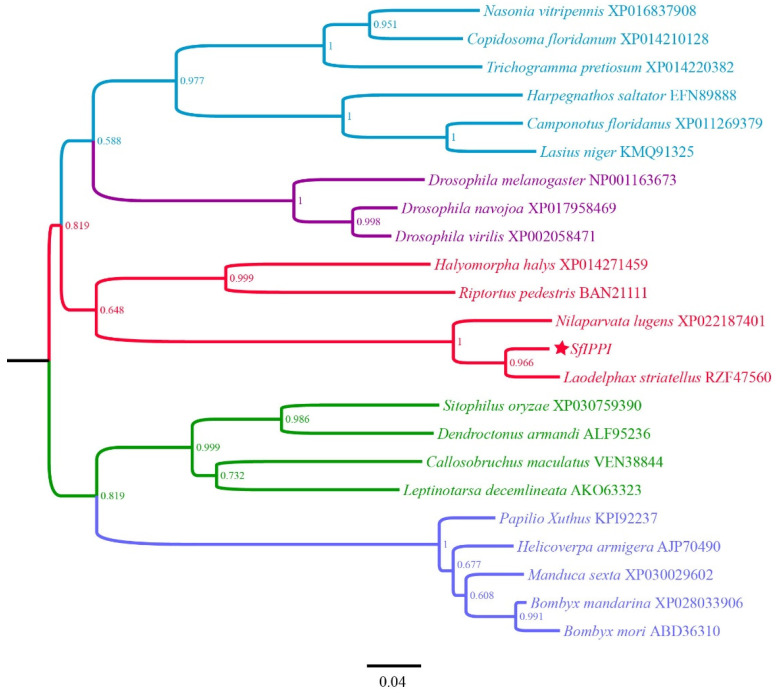
Phylogenetic analysis of *SfIPPI* homologs from insect species based on amino acid sequences. Sequences were downloaded from the GenBank protein database. The red star indicates the *IPPI* gene of *S. furcifera*.

**Figure 3 insects-13-00174-f003:**
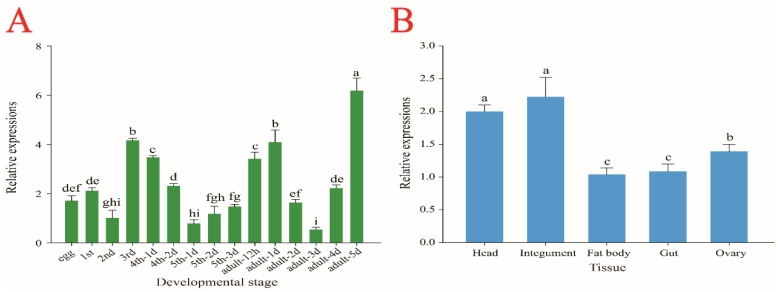
Relative expression levels of *SfIPPI* in the different development stages and tissues of *S. furcifera*. *SfRPL9* and *SfTUB* were used as internal control genes. (**A**) Relative expression levels of *SfIPPI* from egg to 5 d old adult as determined by qPCR. (**B**) Relative expression levels of *SfIPPI* in different tissues of 24 h old female adults as determined by qPCR. The results were expressed as the mean ± standard error of the mean of three biological replicates and their respective three technical replicates. Different lowercase letters above bars indicate significant differences in the gene expression level between different developmental stages and tissues (*p* < 0.05, Tukey).

**Figure 4 insects-13-00174-f004:**
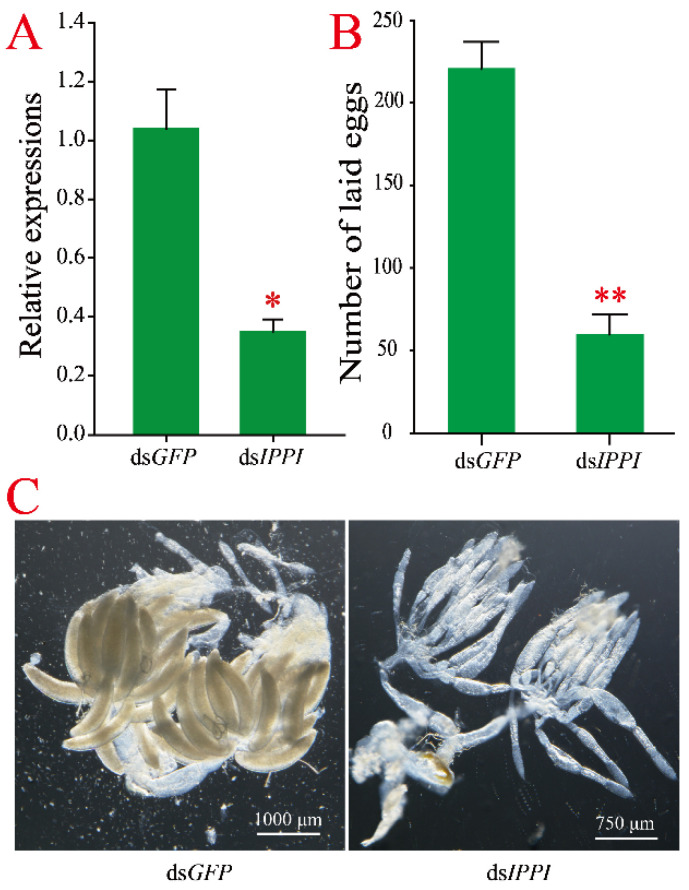
Comparisons of the *S. furcifera* treatment group (ds*IPPI*) and control group (ds*GFP*). (**A**) Effects of *SfIPPI* silencing at the transcription level; (**B**) effects of *SfIPPI* silencing on reproduction; (**C**) effects of *SfIPPI* silencing on ovary development. Significant differences between the treatment and the control groups are indicated by asterisks (* *p*
*<* 0.05 and ** *p*
*<* 0.01).

**Figure 5 insects-13-00174-f005:**
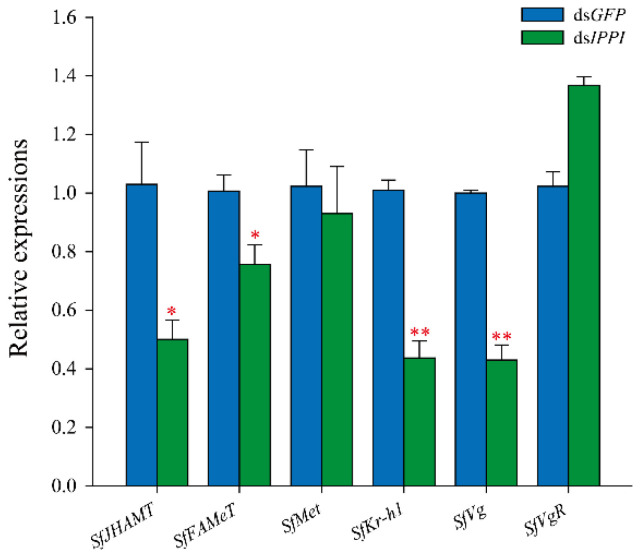
Effect of *SfIPPI* silencing on the expression of other genes in the juvenile hormone signal transduction pathway. Significance of differences between the treatment group (ds*IPPI*) and the control group (ds*GFP*) was determined using the Student *t*-test for independent samples (* *p* < 0.05 and ** *p* < 0.01).

**Table 1 insects-13-00174-t001:** Primers used in this research.

Experiment	Primer	Primer Sequence (5′ to 3′)
cDNA cloning	IPPI-i-F	ATTTGGTAGCAGAGCCATAAGA
	IPPI-i-R	CTCCTGGTTCGCTTCAATT
qPCR	Y-IPPI-F	GCCTGTTGCAGTCATCCGTTGT
	Y-IPPI-R	GCGGTATGCCTAGCTCGTGGTT
	Y-Vg-F	AGTGGTGAGGTGCGTGGTCT
	Y-Vg-R	CGTTGCTGCTGCTACCTGACA
	Y-VgR-F	CTGCGAACACAGCCGAATGGA
	Y-VgR-R	GGAACTGCGACTGCGTATCACA
	Y-JHAMT-F	ACGAGAACCGTAATGGCAGTCA
	Y-JHAMT-R	CCAGGACCACATCCAACATCCA
	Y-FAMeT-F	CTCTTGAACTGACGACCGAGGA
	Y-FAMeT-R	CGACCAGCCGCCTATGAAGAT
	Y-Met-F	GCCGCCAGTTGACCGATTACA
	Y-Met-R	ACCAGCAGAGTCGCACGAGT
	Y-Kr-h1-F	CTCACCGCAGCACTCAACTCA
	Y-Kr-h1-R	AGGCACAGGCGACATTAGAACA
	Y-RPL9-F	GGGCGAGAAGTACATCCGTAGG
	Y-RPL9-R	GCGGCTGATCGTGAGACATCTT
	Y-TUB-F	CGCTGTTGATGGAGAGGCTGTC
	Y-TUB-R	ACGACGGCTGTGGATACCTGTG
dsRNA synthesis	T7-IPPI-F	TAATACGACTCACTATAGGGTAGCAGAGCCATAAGAAGTT
	T7-IPPI-R	TAATACGACTCACTATAGGGGCAGGATTAGAATGTAGTCG
	T7-GFP-F	TAATACGACTCACTATAGGGGCCAACACTTGTCACTACTT
	T7-GFP-R	TAATACGACTCACTATAGGGGGAGTATTTTGTTGATAATGGTCG

Note: Underlined nucleotides indicate DNA sequences transcribed downstream of the T7 promoter. RT-qPCR, reverse transcription real-time polymerase chain reaction; ds RNA, double-stranded RNA.

## Data Availability

All data are provided within the text.

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
