# Peer review of "Juvenile Hormone Synthesis Pathway Gene SfIPPI Regulates Sogatella furcifera Reproduction"

_insects, 2022, doi:10.3390/insects13020174_

Round 1
Reviewer 1 Report
The authors have cloned S. furcifera IPPI, analyzed its expression pattern and studied its function by RNAi. They found RNAi of SfIPPI significantly inhibited the ovarian development and fecundity of females and significantly inhibited the transcription level of downstream genes in the JH synthesis pathway. The paper is very interesting and obviously has merit for publication. However, there are several issues to be addressed before accepting for publication.
L84 Please specify the number collected each developmental stage.
P201 Please cite the reference of 2−ΔΔCt method.
L246-L248 Please make it clear: “midgut” or “gut”?
L256 Please re-write the sub-title.
L274-L279 JHAMT and FAMeT are actually enzymes involved in JH synthesis. Vitellogenin is the precursor of yolk proteins. Please address why these genes were selected for expression level measurement.
L333-335 Please re-write to make the logic “IPPI-JH-Ovary development and reproduction” more clear.
Reviewer 2 Report
Gong et al.: Juvenile Hormone Synthesis Pathway Gene SfIPPI Regulates Sogatella furcifera Reproduction
This is a precious and well-written paper. It will be of interest to the wide community of physicians and biomedical scientists. I have only a couple of minor comments, and I believe the paper can be published almost as it stands.
Lines 40-42: It is crucial to mention the immune function among other physiological processes that are under the control of JH. I would suggest citing a recent paper that appeared in another journal of MDPI: Rantala et al. 2021 J Fungi, doi.org/10.3390/jof6040298
Line 36: Please, avoid using the abbreviated form of isopentenyl pyrophosphate isomerase in the keyword line. Better use either the full name or the full name and IPPI in parentheses. IPPI may confuse the reader.
Line 54: Remove “even.”
Line 244: Figure 3A?
Lines 250-251: You use (a) and (b) in the Figure legend, while A and B are used in the Figure. Please correct for consistency!
The same applies to Figure 4 and its legend.
Line 288: You can remove “Then,”
Line 326: Replace “We showed” with “In this study, we show.”
Lines 337 and 340: “Next” and “Then” can be deleted because your language is fluent enough without these additions.
T. Krama
